# Drought and intimate partner violence towards women in 19 countries in sub-Saharan Africa during 2011-2018: A population-based study

**Adrienne Epstein**[1]*, **Eran Bendavid**[2], **Denis Nash**[3], **Edwin D. Charlebois**[4], **Sheri D. Weiser**[4]

1 Department of Epidemiology and Biostatistics, University of California, San Francisco, San Francisco, California, United States of America, 2 Department of Medicine, Stanford University, Stanford, California, United States of America, 3 Institute for Implementation Science in Population Health, City University of New York, New York, New York, United States of America, 4 Department of Medicine, University of California, San Francisco, San Francisco, California, United States of America

* adrienne.epstein@ucsf.edu

**Data Availability Statement:** Survey data can be accessed through the following website by creating

## Abstract

### Background

Drought has many known deleterious impacts on human health, but little is known about the relationship between drought and intimate partner violence (IPV). We aimed to evaluate this relationship and to assess effect heterogeneity between population subgroups among women in 19 sub-Saharan African countries.

### Methods and findings

We used data from 19 Demographic and Health Surveys from 2011 to 2018 including 83,990 partnered women aged 15–49 years. Deviations in rainfall in the year before the survey date were measured relative to the 29 previous years using Climate Hazards Group InfraRed Precipitation with Station data, with recent drought classified as ordinal categorical variable (severe: ≤10th percentile; mild/moderate: >10th percentile to ≤30th percentile; none: >30th percentile). We considered 4 IPV-related outcomes: reporting a controlling partner (a risk factor for IPV) and experiencing emotional violence, physical violence, or sexual violence in the 12 months prior to survey. Logistic regression was used to estimate marginal risk differences (RDs). We evaluated the presence of effect heterogeneity by age group and employment status. Of the 83,990 women included in the analytic sample, 10.7% (9,019) experienced severe drought and 23.4% (19,639) experienced mild/moderate drought in the year prior to the survey, with substantial heterogeneity across countries. The mean age of respondents was 30.8 years (standard deviation 8.2). The majority of women lived in rural areas (66.3%) and were married (73.3%), while less than half (42.6%) were literate. Women living in severe drought had higher risk of reporting a controlling partner (marginal RD in percentage points = 3.0, 95% CI 1.3, 4.6; p < 0.001), experiencing physical violence (marginal RD = 0.8, 95% CI 0.1, 1.5; p = 0.019), and experiencing sexual violence (marginal RD = 1.2,

an account and filling out a brief form describing intended analyses: https://dhsprogram.com/data/.

**Funding:** This work was supported by National Institutes of Health/National Institute of Allergy and Infectious Disease K24 AI134326-01 (to SDW). The funders had no role in study design, data collection and analysis, decision to publish, or preparation of the manuscript.

**Competing interests:** I have read the journal's policy and the authors of this manuscript have the following competing interests: SDW is a member of the Editorial Board of *PLOS Medicine*. EDC receives grants from NIH.

**Abbreviations:** CHIRPS, Climate Hazards Group InfraRed Precipitation with Station; DHS, Demographic and Health Surveys; EA, enumeration area; IPV, intimate partner violence; RD, risk difference; RERI, relative excess risk due to interaction.

95% CI 0.4, 2.0; $p = 0.001$) compared with women not experiencing drought. Women living in mild/moderate drought had higher risk of reporting physical (marginal RD = 0.7, 95% CI 0.2, 1.1; $p = 0.003$) and sexual violence (marginal RD = 0.7, 95% CI 0.3, 1.2; $p = 0.001$) compared with those not living in drought. We did not find evidence for an association between drought and emotional violence. In analyses stratified by country, we found 3 settings where drought was protective for at least 1 measure of IPV: Namibia, Tanzania, and Uganda. We found evidence for effect heterogeneity (additive interaction) for the association between drought and younger age and between drought and employment status, with stronger associations between drought and IPV among adolescent girls and unemployed women. This study is limited by its lack of measured hypothesized mediating variables linking drought and IPV, prohibiting a formal mediation analysis. Additional limitations include the potential for bias due to residual confounding and potential non-differential misclassification of the outcome measures leading to an attenuation of observed associations.

## Conclusions

Our findings indicate that drought was associated with measures of IPV towards women, with larger positive associations among adolescent girls and unemployed women. There was heterogeneity in these associations across countries. Weather shocks may exacerbate vulnerabilities among women in sub-Saharan Africa. Future work should further evaluate potential mechanisms driving these relationships.

## Author summary

### Why was this study done?

- Extreme weather events (including droughts) are associated with many poor health consequences, yet the link between drought and intimate partner violence has not been studied.

- Previous work has shown that drought is a predictor of many risk factors for intimate partner violence towards women, including food insecurity, migration, and poverty.

### What did the researchers do and find?

- We combined survey data from 19 countries in sub-Saharan Africa with publicly available historical rainfall data to estimate exposure to drought among 83,990 married or partnered women aged 15–49 years, and estimated the association between drought and 4 outcomes related to intimate partner violence.

- Drought was associated with reporting a controlling partner and experiencing physical and sexual violence, with stronger associations among adolescent girls and unemployed women. Drought was not associated with reported emotional violence.

- There was heterogeneity in findings across countries; drought was protective for at least 1 type of violence in Uganda, Namibia, and Tanzania.

**What do these findings mean?**

- Intimate partner violence towards women is yet another potential downstream consequence of the growing intensity and duration of droughts across sub-Saharan Africa.

- Future work should evaluate the pathways linking drought and intimate partner violence in order to best tailor interventions aimed at mitigating drought's impacts.

## Introduction

Climate change, extreme weather events, and in particular droughts negatively impact human health [1–3]. Droughts are becoming increasingly common and of higher intensity in many regions across the globe including sub-Saharan Africa, due, at least in part, to human activity [4,5]. In 2011–2012, East Africa experienced a severe drought that spawned a subsequent refugee crisis [6]. In 2014–2016, Southern Africa experienced 2 years of an El Niño–induced drought, leading to national emergency declarations in a number of countries and exposing 38 million people across the region to drought [7]. In 2019, the number of individuals exposed to severe drought in sub-Saharan Africa swelled to 45 million [8]. There are significant gaps in our understanding of the relationship between droughts and human health.

Droughts lead to reduced agricultural production, impacting households' food security and income [3,9]. These impacts may affect downstream health and behavioral outcomes such as nutrition [2,10], HIV prevalence and risk [11,12], and the incidence of infectious diseases such as diarrhea and respiratory infection [13,14]. One potential downstream effect of drought that has received little attention is intimate partner violence (IPV) towards women. In addition to being a global human rights concern, IPV is associated with a multitude of negative health consequences such as physical injury, high-risk sexual behaviors, HIV, reproductive disorders, adverse pregnancy outcomes, and psychological effects including suicidal ideation, drug abuse, and mental health disorders [15–20]. IPV is especially prevalent in Africa, where 36.6% of ever-partnered women experience physical and/or sexual IPV in their lifetime [21]. Studies suggest that deviations from long-term trends of temperature and precipitation, including drought, are associated with violence at the interpersonal, intergroup, and institutional levels [22–24]. Hypothesized mechanisms include food insecurity and migration [25–27]. There is no literature to date to our knowledge examining associations between drought and IPV. In addition to the above mechanisms, IPV may be influenced by drought through additional pathways including increased inequalities in access to resources, disordered urbanization, poverty, disempowerment, and psychological distress [22,28–30].

Drought may have heterogenous effects on IPV between population subgroups. For example, adolescent girls are at higher risk for IPV globally [31,32] due to their young age and inexperience with relationships. This vulnerability may be exacerbated during times of income instability and food insecurity. Employed women are at lower risk for IPV [33]. They may, in addition, experience fewer negative effects of drought due to economic independence and empowerment, which could lessen the negative impacts of drought on poverty and food insecurity, particularly if employment is not negatively impacted by drought. However, there is no research to our knowledge on the impacts of drought on IPV in these population subgroups.

In the present analysis, we used nationally representative, cross-sectional survey data from 19 countries in sub-Saharan Africa from the period 2011–2018 to investigate the relationships

between drought and emotional, physical, and sexual violence among partnered women. Furthermore, we evaluated the evidence for differences in these associations between population subgroups (age group and employment status).

## Methods

This study is reported following the Strengthening the Reporting of Observational Studies in Epidemiology (STROBE) guidelines (S1 Checklist).

### Data source and participants

This study used data from the Demographic and Health Surveys (DHS), which are cross-sectional, nationally representative household-based surveys funded by the United States Agency for International Development. The surveys use a stratified 2-stage cluster sampling design, selecting first a random sample of enumeration areas (EAs), followed by a random sample of households within each EA. All women aged 15 to 49 years within selected households are invited to complete a questionnaire. A random subset of women in the DHS samples are selected to participate in the domestic violence module.

We used surveys that included geolocated information on each EA and took place during or after 2011, using this year as a cutoff due to the availability of precipitation data (see S1 Table for a full list of surveys included in this analysis). We restricted our sample to the women who participated in the domestic violence module. We excluded women who were not currently married or residing with a male partner. Full information on the outcomes of interest and covariates was also required for inclusion. Mali was excluded from the analysis because no women experienced any level of drought in the study window.

### Measures

**Drought.** Drought was measured using Climate Hazards Group InfraRed Precipitation with Station (CHIRPS) data, which combine satellite imagery with weather station data to create raster rainfall estimates in millimeters at 0.05-decimal-degree resolution from 1981 onward [34]. Annual cumulative precipitation for the 12 months preceding the survey date was calculated for each unique survey date/EA combination. We then ranked this quantity of precipitation with the prior 29 years and converted this ranking to a percentile; for example, the 50th percentile signifies the median level of precipitation in the 30-year period. This use of deviations from long-term precipitation trends is standard in the literature, as it captures weather shocks representing deviations from the norm and therefore does not reflect inherent differences in populations that live in drier or wetter areas [11,12]. We generated an ordinal categorical variable of drought, classifying severe drought as annual precipitation equal to or lower than the 10th percentile of the historical record, mild/moderate drought as above the 10th percentile and equal to or below the 30th percentile, and no drought as above the 30th percentile. We also considered a binary categorization of drought, where drought was defined as annual precipitation in the 12 months prior to the survey that was lower than the 15th percentile of the historical record, reflecting the level of precipitation that impacts GDP and agricultural productivity, as defined previously in the literature [11,12,35]. The calculation and classifications of drought were specified a priori; however, we did not have a written prespecified protocol.

**IPV outcomes.** We considered 4 binary outcomes selected prior to analysis, each representing a different dimension of risk of and/or experience of IPV (see S2 Table for the definitions of outcomes): a binary indicator representing whether the respondent reported a controlling partner (a risk factor for IPV) and binary indicators for whether the respondent

had ever experienced emotional violence, physical violence, or sexual violence by her husband/partner in the 12 months prior to the survey date. We also considered a count variable representing the number of IPV outcomes the respondent endorsed during the survey.

**Covariates.** We included several sociodemographic variables a priori that have theoretical or empirical associations with IPV [36]. These include respondent's age (categorized into 15–19, 20–29, 30–39, and 40–49 years), a binary indicator of respondent literacy (literate versus not literate), a binary indicator of married, the number of live births the respondent has had (categorized into 0, 1–2, 3–4, and 5+), household size (categorized into 2–3, 4–5, and 6+), a binary indicator of rural residence, husband/partner's education level (categorized into none, primary, secondary, or higher), and husband/partner's age (categorized into 15–19, 20–29, 30–39, 40–49, and 50+ years).

**Effect modifiers.** We assessed effect modification by the respondent's age group (adolescent [15–19 years] versus adult [20+ years]) and by the respondent's employment status (binary yes/no for whether the respondent has worked in the last 12 months) to determine whether the associations differed between population subgroups, with the hypothesis that drought would have greater negative consequences for adolescent girls and unemployed women. We also assessed the presence of effect modification by country.

## Statistical analysis

To assess the association between drought and binary IPV outcomes, a series of multivariable logistic regression models were specified for each of the 4 outcomes. For the count outcome (number of IPV outcomes endorsed), we specified an ordered logistic regression model. In all models, we included country fixed effects to control for country-level differences, such as in norms, sociodemographic characteristics, and economies, and we included robust standard errors clustered at the EA level. Because we included country fixed effects, our models are "within" estimators, comparing women with different drought statuses within a given country. We first ran the models with only country-level fixed effects and the drought variable (categorical and binary classifications included separately), and subsequently added in the covariates. We then derived marginal risk differences (RDs) by implementing Stata's *margins* command, which estimates effects by first deriving predictions for each observation in the sample as if they were in each level of the exposure (no drought, mild/moderate drought, and severe drought). For each level of exposure, each individual's marginal effect is then estimated by subtracting the predicted probability of the outcome under the reference group condition (no drought) from the predicted probability under drought conditions. The effect estimate is then averaged across all observations.

To assess whether our results were sensitive to country-specific outliers, we conducted sensitivity analyses alternately leaving out each of the 19 countries in the pooled models. To assess the presence of effect modification, we generated interaction terms between hypothesized effect modifiers and the binary drought variable; we then included the interaction terms and main effects in models for each outcome. We assessed interaction on the additive scale by calculating the relative excess risk due to interaction (RERI) [37]. We considered an alpha significance level of 0.10 for the RERI term. All analyses were carried out in Stata 14 and R version 3.4.

## Ethical approval

DHS obtains informed and voluntary consent from survey participants, and permission to use DHS data was obtained from the DHS program. Specific approval for this de-identified secondary data analysis was not required.

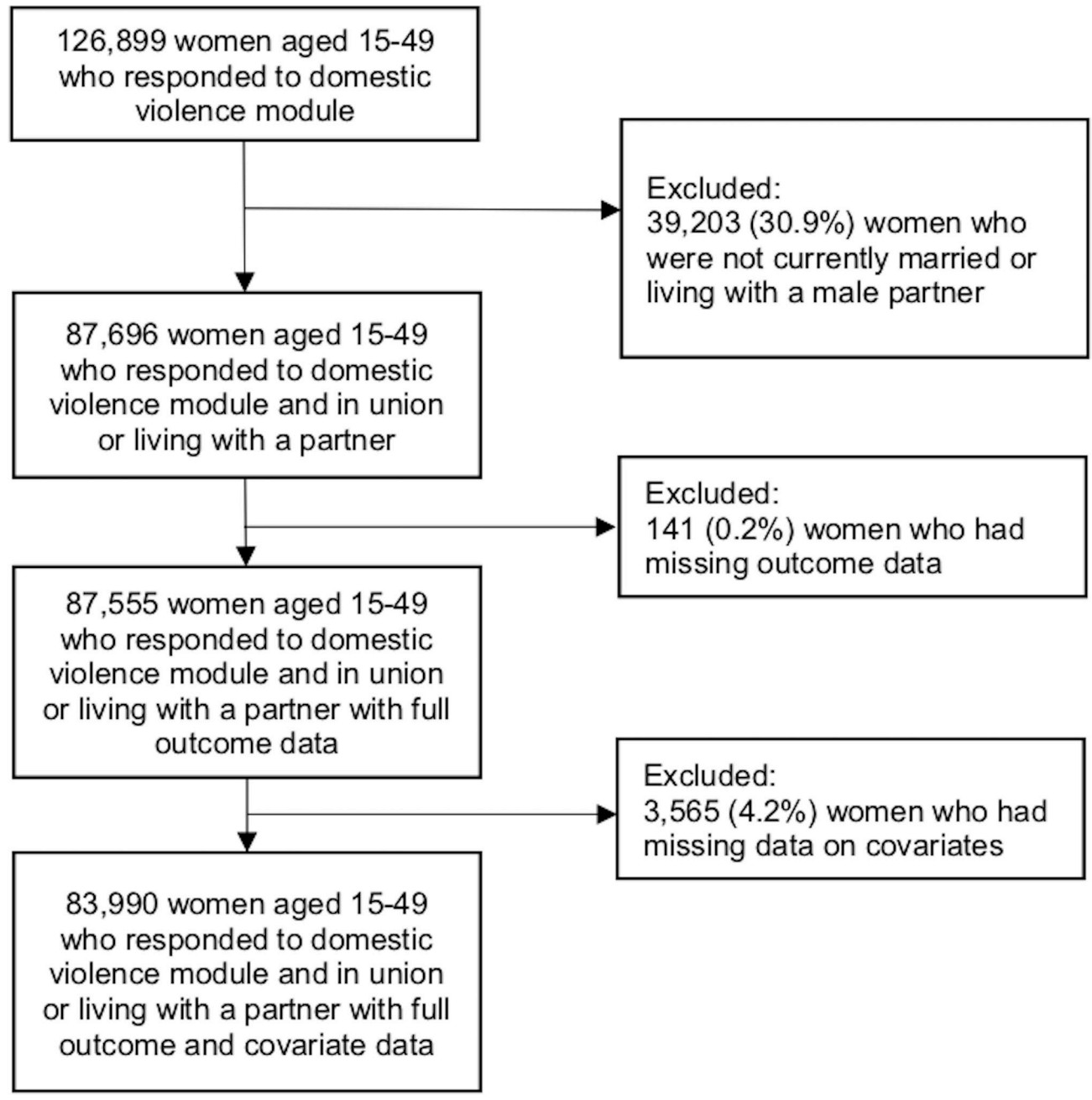

**Fig 1. Flow chart depicting how the final analytic sample was selected.**

## Results

The analytic sample included 83,990 eligible women from 19 countries in sub-Saharan Africa. See Fig 1 for how the sample size was determined and S1 Table for the sample size in each country.

Table 1 shows the outcomes and sociodemographic characteristics among respondents. In sum, under half (42.6%) of respondents were literate, and nearly three-quarters (73.3%) were married. A majority (66.3%) of the sample resided in rural areas. Most women (93.7%) had

**Table 1. Descriptive statistics of women aged 15 to 49 years currently married or living with a male partner included in IPV analyses (*n* = 83,990).**

| Covariate or outcome | Number (percent) (*n* = 83,990) |
|---|---|
| Age category (years) | |
| 15–19 | 5,316 (6.3) |
| 20–29 | 35,036 (41.7) |
| 30–39 | 29,052 (34.6) |
| 40–49 | 14,586 (17.4) |
| Literate | 35,817 (42.6) |
| Married | 61,569 (73.3) |
| Employed within previous 12 months | 61,666 (73.4) |
| Number of births | |
| 0 | 5,316 (6.3) |
| 1–2 | 26,876 (32.0) |
| 3–4 | 24,947 (29.7) |
| 5+ | 27,329 (32.5) |
| Household size | |
| 2–3 | 17,698 (21.1) |
| 4–5 | 29,311 (34.9) |
| 6+ | 36,981 (44.0) |
| Rural residence | 55,717 (66.3) |
| Husband/partner's education | |
| No education | 18,993 (22.6) |
| Primary | 31,987 (38.1) |
| Secondary | 27,441 (32.7) |
| Higher | 5,569 (6.6) |
| Husband/partner's age category (years) | |
| 15–19 | 483 (0.6) |
| 20–29 | 18,674 (22.2) |
| 30–39 | 31,706 (37.8) |
| 40–49 | 21,171 (25.2) |
| 50+ | 11,956 (14.2) |
| IPV outcomes | |
| Reported a controlling partner | 55,628 (66.2) |
| Ever experienced emotional violence in previous 12 months | 15,992 (19.0) |
| Ever experienced physical violence in previous 12 months | 4,390 (5.2) |
| Ever experienced sexual violence in previous 12 months | 3,539 (4.2) |

IPV, intimate partner violence.

had at least 1 live birth prior to the survey. All age categories were represented, with the lowest proportion in the adolescent (15–19 years) category (6.3%). The IPV outcomes were common: 66.2% of women reported a controlling partner, and 19.0%, 5.2%, and 4.2% of women reported having experienced emotional, physical, and sexual violence in the 12 months prior the survey, respectively. The distribution of drought status differed substantially across countries (Fig 2), ranging from 2.9% of respondents experiencing any form of drought in Cameroon to 95.6% in Togo.

Table 2 shows the associations between drought and the IPV outcomes, with drought categorized as mild/moderate drought and severe drought compared with no drought. In adjusted analyses, women living in severe drought had a 3-percentage-point higher risk of reporting a

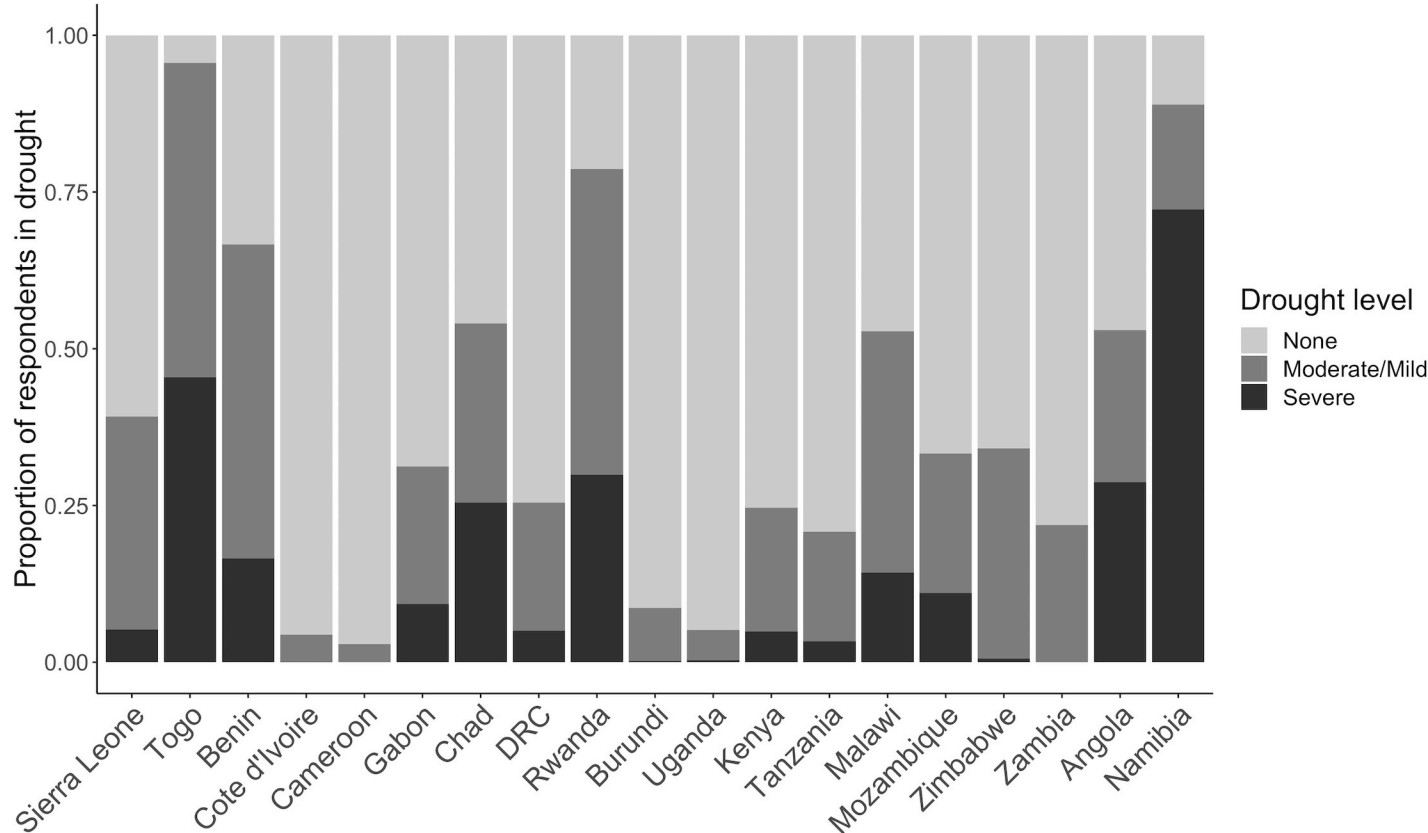

**Fig 2. Proportion of respondents experiencing drought over the past 12 months in each country included in the analysis.** DRC, Democratic Republic of the Congo.

controlling partner (marginal RD = 3.0, 95% CI 1.3, 4.6; $p < 0.001$) compared to those not experiencing drought. Mild/moderate drought was not associated with reporting a controlling partner (marginal RD = 0.0, 95% CI −1.1, 1.2; $p = 0.95$). Severe and mild/moderate drought were positively associated with reported physical violence (marginal RD for severe drought = 0.8, 95% CI 0.1, 1.5; $p = 0.019$; marginal RD for mild/moderate drought = 0.7, 95% CI 0.2, 1.1;

**Table 2. Associations between severe and mild/moderate drought and intimate partner violence among women aged 15–49 years ($n = 83,990$).**

| Exposure | Outcome | | | | | | | |
|---|---|---|---|---|---|---|---|---|
| | Reported a controlling partner | | Emotional violence in previous 12 months | | Physical violence in previous 12 months | | Sexual violence in previous 12 months | |
| | Unadjusted | Adjusted | Unadjusted | Adjusted | Unadjusted | Adjusted | Unadjusted | Adjusted |
| No drought | REF | REF | REF | REF | REF | REF | REF | REF |
| Mild/moderate drought | 0.3 (−0.8, 1.2) | 0.0 (−1.1, 1.2) | 0.4 (−0.5, 1.3) | 0.4 (−0.1, 1.7) | 0.7** (0.3, 1.2) | 0.7** (0.2, 1.1) | 0.7** (0.3, 1.2) | 0.7** (0.3, 1.2) |
| Severe drought | 3.0*** (1.4, 4.7) | 3.0*** (1.3, 4.6) | 0.6 (−0.8, 1.9) | 0.4 (−0.5, 1.3) | 0.9* (0.2, 1.6) | 0.8* (0.1, 1.5) | 1.3** (0.5, 1.2) | 1.2** (0.4, 2.0) |

Coefficients are presented as marginal risk difference estimates in percentage points from logistic regression models, with 95% confidence intervals in parentheses. The unadjusted model includes country-level fixed effects. The adjusted model includes age category, literacy, marital status, number of births, household size, rural residence, husband/partner's age, and husband/partner's education. Standard errors are clustered at the enumeration area level. Asterisks denote level of significance

*$p < 0.05$

**$p < 0.01$

***$p < 0.001$.

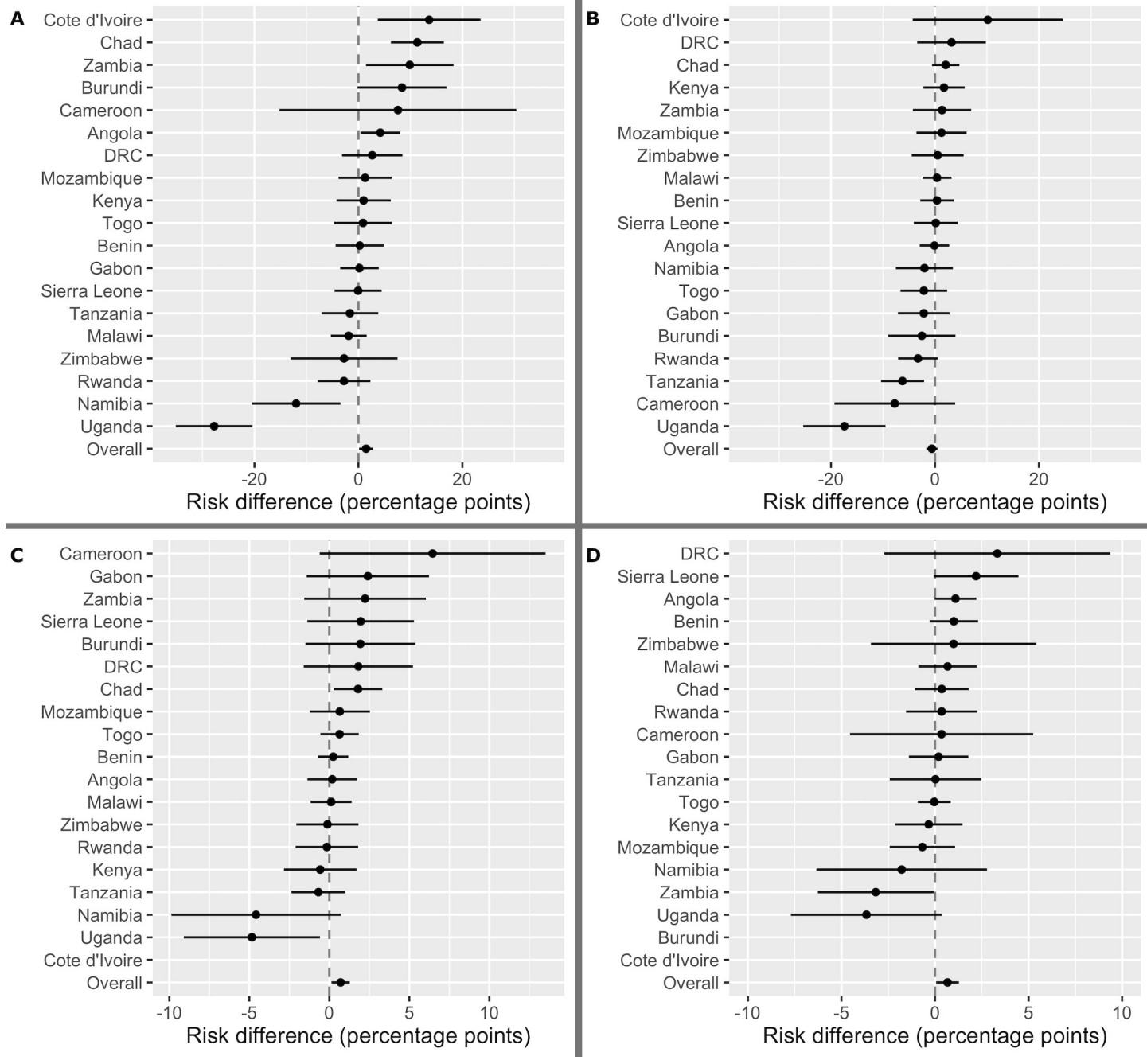

**Fig 3. Country-specific adjusted associations between drought and intimate partner violence.** Country-specific adjusted associations between drought and (A) reporting a controlling partner, (B) experiencing any emotional violence in the prior 12 months, (C) experiencing any physical violence in the prior 12 months, and (D) experiencing any sexual violence in the prior 12 months. All models control for respondent age category, literacy, marital status, number of births, household size, rural residence, husband/partner's age, and husband/partner's education. Associations are presented as risk differences (percentage points) and 95% confidence intervals. Standard errors are clustered at the enumeration area level. Results not shown for countries with insufficient outcome data. DRC, Democratic Republic of the Congo.

$p$ = 0.003) and sexual violence (marginal RD for severe drought = 1.2, 95% CI 0.4, 2.0; $p$ = 0.001; marginal RD for mild/moderate drought = 0.7, 95% CI 0.3, 1.2; $p$ = 0.001). We did not find evidence for associations between drought and emotional violence. These results were consistent when drought was classified as a binary variable (S3 Table). Associations between

**Table 3. Associations between drought and intimate partner violence among women aged 15–49 years stratified by age and employment status ($n$ = 83,990).**

| Outcome | Risk difference (95% CI) | | | |
| --- | --- | --- | --- | --- |
| | Stratified by age category | | Stratified by employment status | |
| | 15–19 years | 20+ years | Not employed | Employed |
| Reported a controlling partner | 4.4* (0.9, 7.9) | 1.3 (0.0, 2.6) | 3.2** (1.1, 5.4) | 0.7 (−0.9, 2.2) |
| Ever experienced emotional violence in previous 12 months | 3.2* (0.1, 6.3) | −0.8 (−1.9, 0.3) | 0.3 (−1.3, 1.9) | −0.8 (−2.2, 0.5) |
| Ever experienced physical violence in previous 12 months | 2.0* (0.1, 3.8) | 0.6* (0.0, 1.2) | 1.1** (0.2, 2.0) | 0.3 (−0.3, 1.0) |
| Ever experienced sexual violence in previous 12 months | −0.2 (−1.9, 1.6) | 0.7* (0.1, 1.4) | 1.5** (0.4, 2.5) | 0.1 (−0.6, 0.9) |

Coefficients are presented as risk difference estimates in percentage points from logistic regression models, with 95% confidence intervals in parentheses. Adjusted for respondent age category, literacy, marital status, number of births, household size, rural residence, husband/partner's age, and husband/partner's education. Standard errors are clustered at the enumeration area level. Asterisks denote level of significance

*$p < 0.05$

**$p < 0.01$.

drought and IPV in addition to all covariates are shown as odds ratios in S4 Table. Models assessing the relationship between drought and a count of IPV outcomes endorsed showed similar results, with a positive, significant association between severe drought and IPV and a marginally positive, significant association between mild/moderate drought and IPV (S5 Table).

Fig 3 shows the associations between drought and the 4 IPV outcomes as marginal RDs for each country individually, with drought classified as a binary variable (<15th percentile of mean annual precipitation). Five countries demonstrated positive, significant associations between drought and reporting a controlling partner; 1 country demonstrated a positive, significant association between drought and physical violence; and 2 countries demonstrated positive, significant associations between drought and sexual violence. These country-level analyses also revealed scenarios in which drought was protective against IPV. In Uganda, drought was protective for all outcomes, although we could not rule out a null or positive association between drought and reported sexual violence. In Namibia, drought was protective for reporting a controlling partner. Finally, in Tanzania, drought was protective for reported emotional violence. The results were significantly heterogeneous between countries for reporting a controlling partner ($p$ for joint interaction term < 0.001), emotional violence ($p$ for joint interaction term = 0.009), and physical violence ($p$ for joint interaction term = 0.022), but not for reported sexual violence ($p$ for joint interaction term = 0.40). Due to the heterogeneity of these associations, we specified pooled models sequentially leaving out 1 country; pooled results remained consistent for all outcomes (S6 Table). We also specified pooled models using mixed effects logistic regression (with random intercepts at the country level) in order to account for cluster heterogeneity; results were qualitatively consistent (in magnitude, direction, and statistical significance) across specifications (S7 Table).

We found evidence for additive effect modification of the association between age (adolescent versus adult) and drought and between employment status and drought. RDs stratified by age category and employment status are presented in Table 3. Adolescents who experienced drought had a higher risk of reporting a controlling partner (marginal RD = 4.4, 95% CI 0.9, 7.9; $p$ = 0.011) and emotional violence (marginal RD = 3.2, 95% CI 0.1, 6.3; $p$ = 0.030) than those not living in drought. Among adult women, drought was not associated with reporting a controlling partner (marginal RD = 1.3, 95% CI 0.0, 2.6; $p$ = 0.063) nor with emotional violence (marginal RD = −0.8, 95% CI −1.9, 0.3; $p$ = 0.14). This heterogeneity was statistically significant (drought–adolescent RERI $p$ = 0.092 for reporting a controlling partner and drought–

adolescent RERI $p = 0.011$ for emotional violence). Drought was positively associated with risk of reported physical violence in both age groups, with no evidence for effect heterogeneity (drought–adolescent RERI $p = 0.16$). Drought was also associated with sexual violence among adult women (marginal RD = 0.7, 95% CI 0.1, 1.4; $p = 0.014$). This association was not statistically significant among adolescent girls (marginal RD = −0.2, 95% CI −1.9, 1.6; $p = 0.86$); however, there was no evidence for effect heterogeneity (drought–adolescent RERI $p = 0.34$).

There was also evidence of interaction between drought and employment for reporting a controlling partner (drought–employment RERI $p = 0.050$) and reporting sexual violence (drought–employment RERI $p = 0.022$). In both of these instances, unemployed women demonstrated positive associations between drought and IPV, whereas we did not observe associations among employed women. Drought was associated with greater risk of reported physical violence among unemployed women but not employed women, although there was no evidence for effect heterogeneity (drought–employment RERI $p = 0.13$). Drought and emotional violence were not associated among employed nor unemployed women (drought–employment RERI $p = 0.19$)

## Discussion

While previous studies have found associations between heat waves and aggressive or violent behavior [38–40], this study extends the literature on climate variation and violence by exploring the relationship between drought and IPV towards women. Using cross-sectional surveys of 83,990 women from 19 countries in sub-Saharan Africa from the period 2011–2018, we found associations between drought and several manifestations of IPV in pooled analyses. Women in mild/moderate drought were at similar risk of physical violence to those in severe drought, with 0.7-percentage-point and 0.8-percentage-point marginal RDs, respectively. These estimates are large in magnitude given the prevalence of reported physical violence in the sample (5.2%), corresponding to 14% higher risk of reported physical violence in mild/moderate drought and 15% higher risk in severe drought. Women in severe drought were also at greater risk of reported sexual violence than those in mild/moderate drought. Similarly, these marginal RDs are substantially higher than baseline risk: severe drought was associated with a 28.6% higher prevalence of women reporting sexual violence in the sample (4.2%), and mild/moderate drought was associated with a 17% higher prevalence. Women who experienced severe drought were more likely to report having a controlling partner, a risk factor for IPV, while those in mild/moderate drought were not. Finally, we did not find evidence for an association between drought and emotional IPV in the pooled sample.

The findings from this analysis coincide with the broader literature on climate and violence, which suggests that drought is associated with increased conflict at the national and subnational levels [22–24]. These associations are attributed to rising commodity prices and increased scarcity of resources such as fresh water during drought. Drought has also been linked to increases in the incidence of personal violence, such as murder [41] and property crimes [22,42]. Our findings suggest that another manifestation of this relationship between drought and violence may exist in the context of IPV.

There are several potential mechanisms that could explain the relationships between drought and dimensions of IPV. Drought may impact a household's income by negatively affecting agricultural production, food supply, health, and household savings. Poverty, in turn, is associated with IPV [25–27]. These income and food production shocks may lead to food insecurity, which has been linked with IPV in several settings, including Nepal, the United States, Brazil, and Southern Africa [43–47]. Both food insecurity and poverty create risk for IPV through the pathway of stress [48,49], which results from hunger, worry about food

access, and financial strain on the household. Stress may impact physical IPV and could also affect a partner's desire to control his wife's movements and behaviors. Poverty and food insecurity can also lead to poor mental health conditions such as depression, a risk factor for IPV [49,50]. In addition, poverty and food insecurity may lead to disempowerment. For example, living in a food-insecure and impoverished household may impact a woman's ability to leave an abusive partner, due to economic dependence [51]. Finally, drought is a known contributor to migration [52–54], and migrant women are at higher risk for IPV due to their lack of social support and the added vulnerability of their migrant status [55–57].

We found interactions between drought and age, such that adolescent girls in drought demonstrated higher risk of reporting a controlling partner and emotional violence, while adult women did not. Research has found that younger women, including adolescent girls, are at higher risk for IPV [31]. Our findings suggest that the impacts of drought exacerbate these vulnerabilities, potentially due to the fact that younger women have lower social standing and are relatively inexperienced [32]. We did not find evidence for an association between drought and reported sexual violence among adolescent girls; this may be due to study power constraints, given the small number of adolescent girls who experienced sexual violence ($n$ = 247). We also found that unemployed women in drought had higher risk of reporting a controlling partner, sexual violence, and physical violence, while these associations were not found in employed women. Both of these findings—that younger women and unemployed women in drought are at higher risk for IPV—suggest that power imbalances may be compounded by drought and subsequent poverty and food insecurity. These 2 findings may be linked, as younger women are less likely to be economically independent from their partners. In contrast with our findings and other literature documenting a protective association between employment and IPV [33], some previous work has suggested that female employment may increase risk of IPV due to perceived power imbalances by the spouse [58,59]. However, in the context of drought, it may be that the woman's income contributes to household resilience to poverty and food insecurity associated with drought, which in turn decreases risk for violence. Given the cross-sectional nature of the data, it is difficult to disentangle the directionality of the relationship between employment and IPV. Qualitative and longitudinal studies are needed to elucidate the mechanisms underpinning the association between employment and IPV in the context of drought.

Contrary to our hypothesis, we found 3 settings in which drought was protective against IPV. In Namibia, a setting with markedly high prevalence of drought, women who lived in drought areas were less likely to report a controlling partner and physical violence. In Uganda and Tanzania, countries with very low prevalence of drought (7.9% and 0.7%, respectively), we found drought was protective for reporting emotional violence. One potential explanation is that these countries experienced excessive rains during the 30 years prior to the survey, and therefore drought may reflect reprieve. More research, including qualitative studies, is needed in these countries to elucidate potential mechanisms driving these associations and to better understand the heterogeneity in these findings.

We did not find evidence for an association between drought and emotional violence in the pooled sample, also contrary to our hypothesis. This may be because emotional violence is less clearly defined than physical and sexual violence and is therefore more prone to measurement error, leading to an attenuation of the association.

## Strengths and limitations

The strength of this study is that it includes 19 different countries in sub-Saharan Africa, representing varying agricultural systems, environments, and sociodemographic makeups. The

surveys took place across an 8-year window and represent a range of drought conditions. The potential for confounding is low in this study because we defined drought as precipitation relative to the 29 previous years, and, as such, the exposure should be independent from potential confounding variables, that is, we have removed variation representing factors associated with historically drier or wetter places.

This analysis has several limitations. First, causal claims are challenging in this context due to the cross-sectional nature of DHS surveys. However, given the low likelihood of reverse causality, the directionality of this hypothesized relationship is supportable. Second, there may be inconsistencies in how data were collected across countries and years with respect to IPV. However, the DHS program strives to achieve standardization in data collection procedures across locations and timepoints. Third, the IPV outcomes may be misclassified because women tend to underreport experiences of IPV, which could affect the magnitude of associations, and there may be cultural differences in reporting IPV across regions and countries. However, we do not believe that IPV reporting bias would depend on drought exposure status within countries, and therefore any bias would be toward the null on average, suggesting that our results underestimate the magnitude of true associations. Fourth, although we hypothesize several mechanisms, this study does not include a direct mediation analysis. A formal mediation analysis was not possible in this context due to the lack of collection of data in the DHS on several hypothesized mediators. For example, although we have data on wealth measured with an asset index, we do not believe this adequately captures the income and expenditure shocks associated with drought. Repeating this analysis in a new dataset with measures of hypothesized mediators, including mental health, food security, migration history, and expenditure indicators, could be an important next step in this field of research. Fifth, we do not include sampling weights in this analysis, which may limit the generalizability of our findings. However, the inclusion of 19 countries across sub-Saharan Africa, rather than restricting the study to 1 country, enhances the external validity of these findings. Sixth, the IPV questions were only asked of married and cohabiting women, and hence the results are not generalizable to women with a different relationship status. Since unmarried, non-cohabiting women are also at risk for IPV, future studies should assess these associations among all women. Seventh, we were unable to assess the impact of repeated exposure to drought, given the availability of CHIRPS rainfall data. Eighth, given the observational nature of the data, residual confounding may bias the observed associations. However, by using EA-level deviation from long-term precipitation as the definition of drought, the possible variation related to sociodemographic factors associated with historically drier or wetter places that may also impact IPV should be removed, and therefore we do not expect confounding to be substantial. Finally, the CHIRPS precipitation dataset relies on both satellite data and ground stations. The distribution of stations across sub-Saharan Africa is not consistent, and some countries may have less accurate precipitation data than others, leading to the potential for misclassification of drought in some settings. However, by classifying drought as the percentile of annual precipitation over the past 30 years, rather using absolute estimates of rainfall, we believe we have reduced the potential for misclassification.

## Conclusions and implications

There is a growing body of literature on the health effects of drought across the globe. Our study contributes to this field of inquiry, with results suggesting that drought may impact IPV. This finding has broad implications, as drought may be contributing not only to IPV, but to downstream health-related consequences of IPV such as reproductive disorders, physical injury, and psychological effects. Given the anticipated acceleration of weather shocks and

drought events in the coming years, more research is imperative to elucidate the pathways linking drought and violence in order to best tailor interventions aimed at reducing the effects of drought on IPV.

## Supporting information

**S1 Checklist. STROBE checklist.**
(DOC)

**S1 Table. List of surveys included in analysis.**
(DOCX)

**S2 Table. Definition and dimensions of the outcomes considered in this analysis.**
(DOCX)

**S3 Table. Associations between drought and IPV among women aged 15–49 years in pooled analysis with drought considered as a binary variable.**
(DOCX)

**S4 Table. Associations between drought and IPV among all women aged 15–49 years in pooled analysis with covariates.**
(DOCX)

**S5 Table. Associations between drought and number of IPV outcomes endorsed among all women aged 15–49 years in pooled analysis.**
(DOCX)

**S6 Table. Associations between drought and IPV among women aged 15–49 years in pooled analysis with each country sequentially removed.**
(DOCX)

**S7 Table. Associations between drought and IPV among women aged 15–49 years in pooled analysis specified with multilevel logistic regression.**
(DOCX)

## Author Contributions

**Conceptualization:** Adrienne Epstein, Denis Nash, Edwin D. Charlebois, Sheri D. Weiser.

**Formal analysis:** Adrienne Epstein.

**Methodology:** Adrienne Epstein, Eran Bendavid.

**Supervision:** Eran Bendavid, Sheri D. Weiser.

**Writing – original draft:** Adrienne Epstein, Sheri D. Weiser.

**Writing – review & editing:** Adrienne Epstein, Eran Bendavid, Denis Nash, Edwin D. Charlebois, Sheri D. Weiser.

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
