## [Decision Letter · Decision Letter 0]

24 Nov 2019

Dear Dr. Epstein,

Thank you very much for submitting your manuscript "Drought and intimate partner violence: findings from 19 countries in sub-Saharan Africa" (PMEDICINE-D-19-03834) for consideration at PLOS Medicine. 

Your paper was discussed among the editorial team and sent to independent reviewers, including a statistical reviewer. The reviews are appended at the bottom of this email and any accompanying reviewer attachments can be seen via the link below:

[LINK]

In light of these reviews, we will not be able to accept the manuscript for publication in the journal in its current form, but we would like to invite you to submit a revised version that fully addresses the reviewers' and editors' comments. You will appreciate that we cannot make a decision about publication until we have seen the revised manuscript and your response, and we expect to seek re-review by one or more of the reviewers. 

We hope to receive your revised manuscript by Dec 13 2019 11:59PM. Please email us (plosmedicine@plos.org) if you have any questions or concerns.

Please let me know if you have any questions. Otherwise, we look forward to receiving your revised manuscript in due course. 

Sincerely,

Richard Turner, PhD

rturner@plos.org

Please add an additional sentence or two to the data statement so as to assist readers in locating and accessing the study data. 

Please adapt your title so that the portion after the colon makes the study design (e.g. "a cross-sectional study") clear. 

We ask you to note in the "methods and findings" subsection of your abstract the countries which do not fit with the pattern of "drought associated with IPV". 

In your abstract and elsewhere, please include p values alongside CI where available. 

Please add summary demographic details about study participants to your abstract. 

Please add a new final sentence to the "methods and findings" subsection of your abstract, in which you summarize the study's main limitations.

After the abstract, we will need to ask you to add a new and accessible "author summary" section in non-identical prose. You may find it helpful to consult one or two recent research papers published in PLOS Medicine to get a sense of the preferred style. 

At line 48, please add "In this study, we found that ..." or similar.

In the paragraph at lines 85-89, please use a consistent tense (i.e., "we used ... we evaluated").

Early in the methods section of your main text, please state whether or not the study had a protocol or prespecified analysis plan, and if so attach the document(s) as a supplementary file (referred to in the methods section). Please highlight analyses that were not prespecified. 

To your methods section, please add a brief statement on ethics approval, which might state that specific approval was not required for the present study, for example. 

Throughout your text, please format reference call-outs as follows: "... human activity [4,5].".

Please avoid "-0.0", as at line 234, for example. 

Please ensure that all reference citations have full access information (e.g., reference 9). References 29, 32 and 41 may need some additional information. 

Please add a completed checklist for the most appropriate reporting guideline, which may be STROBE or RECORD, referred to in your methods section. In the checklist, individual items should be referred to by section (e.g., "Methods") and paragraph number rather than by page or line numbers, as the latter generally change in the event of publication. 

Comments from the reviewers:

*** Reviewer #1: 

This is a very impressive and important study detailing the social and psychological impact of droughts in Sub-Saharan Africa. The authors have performed a very commendable job of using nationally-representative data from 19 countries and demonstrate that exposure to prolonged periods of drought is associated with different forms of interpersonal violence. It points to the need to address the emotional impacts of a very specific climate-related event which threaten the lives of women in particular and families in general as they struggle to cope with the economic consequences of such events. 

There are a few suggestions offered that would improve the quality and impact of the manuscript, however. First, the authors should probably document the number of people whose lives have been affected by drought in Sub-Saharan Africa, both today and during the study period. Such information is readily available from sources such as the UN and the WHO. Second, IPV was assessed using four binary indicators, but this raises the question as to whether someone could experience more than one form of IPV. It would be helpful if the authors examined whether there was a dose-response relationship between extent of drought and number of IPV indicators or explain why such a comparison is not meaningful or possible with the data. Third, while Figure 2 illustrates differences in drought level by country, there are also important differences in drought level within each country during the study period. Women in some of these countries may have been exposed to longer periods of drought than others but the results do not appear to reflect this. If the ranking of quantity of precipitation does reflect these differences, then the authors should explain how it does so. Fourth, the proposed conceptual framework presented in the discussion section on p. 19 should be eliminated as the study provided no results supporting it. A discussion of potential causal mechanisms is appropriate, but presentation of a conceptual framework such as the one proposed in the manuscript is premature. Fifth, while the inclusion of 19 countries may enhance the external validity of the study findings, it also increases the likelihood of ignoring important cultural differences within those 19 countries. This should be addressed as a potential limitation as cultural differences may result in different patterns of reporting of the different forms of IPV. While drought may reflect a reprieve in Uganda and Tanzania, it does not appear to be the case in Namibia. 

*** Reviewer #2: 

This is a well-written study investigating ecological associations at a country level between drought and IPV in sub-Saharan Africa. The introduction presents a clear justification for the study. The methods are reported in appropriate detail and there are good measures of drought rates and sensitive measures of IPV across 4 outcomes (being controlled, victim of violence, emotional abuse, and sexual violence). These IPV-related outcomes are based on self-report measures and the authors excluded 31% of participants who were asked questions in the domestic violence module of a questionnaire as they were not married or living with a partner. 

Some major areas that need clarification: 

1. Why did the authors exclude these 31% of participants. Is it the case that only those who are married or living with a male partner can experience IPV? I would assume that intimate relationships (and therefore IPV) can exist outside of these inclusion criteria, and thus the IPV estimates reported in the paper may be biased one way or the other due to this exclusion (and I would suspect that they are biased upwards). 

2. The covariates were categorized a priori. Was this the same for other variables (e.g. drought, IPV-related outcomes)? Was there a statistical plan? 

3. The findings appear to show considerable between-country heterogeneity, which would question their decision to pool findings and present overall marginal risk differences. Can they estimate the degree of heterogeneity? Looking at control-related outcome, 11 countries had no clear relationship with drought, 2 were negatively associated, and 6 were positively associated. This is too much statistical heterogeneity in my view for a pooled estimate. 

4. There is also heterogeneity across the 4 IPV-related outcomes - in that emotional outcomes are not associated with drought, unlike pooled estimates for sexual violence or interpersonal violence. 

5. The conclusion starts referring to 'increased risk' whereas the results discussed marginal risk differences. Consistency in the presentation of the findings is required. 

Overall, with the substantial between-country and across-outcome heterogeneities, I think that the findings do not warrant the relatively firm conclusions drawn. Other measures of IPV would be helpful to triangulate the findings, which as they stand are hypothesis-generating. 

*** Reviewer #3: 

Alex McConnachie

This review considers the use of statistics in the paper by Epstein and colleagues, which investigates the association between periods of drought and reports of intimate partner violence in sub-Saharan Africa.

Overall, I think the statistical elements of the paper are very good. The data sources are well described, and the use of logistic regression to assess the association of interest is perfectly reasonable. Presenting the results on an absolute scale, rather than as odds ratios, is an interesting approach, and perfectly reasonable. Use of interactions to assess variations in associations between subgroups is good.

The comments that I have are generally quite minor.

Line 140 mentions the minimum household size as being 1, when I think it should be 2. I saw the same thing in Appendix S4.

Line 199 uses the word "relationship", when "association" is slightly preferable, to avoid any implication of causality.

Figure 3 shows country-specific associations. Are these derived from separate models, with different covariate effects in each model, or from models with country-by-drought interactions? Are these associations significantly different? I suspect they are, but a p-value would help.

Line 291 raised an interesting point. Does this mean that it was not possible from the survey data used, to identify women who had experienced IPV in the previous 12 months, but had left the household by the time of the survey? Even if this were possible, it would probably also be necessary to identify women who had left a partner in the past 12 months without experiencing IPV, and I can imagine this would be more difficult.

***

[LINK]

---

## [Decision Letter · Decision Letter 1]

29 Jan 2020

Dear Dr. Epstein,

Thank you very much for re-submitting your manuscript "Drought and intimate partner violence: a population-based study from 19 countries in sub-Saharan Africa" (PMEDICINE-D-19-03834R1) for consideration at PLOS Medicine.

I have discussed the paper with editorial colleagues and it was also seen again by two reviewers. I am pleased to tell you that, provided the remaining editorial and production issues are dealt with, we expect to be able to accept the paper for publication in the journal.

[LINK]

We hope to receive your revised manuscript within one week. Please email us (plosmedicine@plos.org) if you have any questions or concerns.

Please let me know if you have any questions. Otherwise, we look forward to receiving the revised manuscript shortly. 

Sincerely,

Richard Turner, PhD

rturner@plos.org

Requests from Editors:

Please add to your conflict of interest statement that SDW is a member of PLOS Medicine's editorial board.

We ask you to restructure the title to better match journal style, and suggest "Drought and intimate partner violence in 19 countries in sub-Saharan Africa during 2011-2018: a population-based study".

Can you quote a mean age for study participants around line 40?

At line 65, we suggest amending the text to "... intimate partner violence towards women." or similar. We suggest making similar amendments elsewhere in your ms, e.g. at line 80 and in the title.

To the sentence at the end of the "methods and findings" subsection of your abstract summarizing study limitations, we ask you to add a few words to mention 1-2 further limitations, such as the possibility of unmeasured confounding affecting the results. 

Around line 162 of your main text, please state explicitly that your study did not have a written prespecified analysis plan or protocol. 

Around line 393, we suggest mentioning possible umeasured confounding in the list of potential study limitations.

Please add an author or group name to reference 8.

Please add additional access information to references 1 and 37 as needed. 

Noting references 29 and 30, for example, please ensure that journal names are abbreviated as appropriate in your reference list. 

Comments from Reviewers:

*** Reviewer #1: 

The revised manuscript appears somewhat responsive to the reviews. The authors appear to have misunderstood recommendation 2 provided by Reviewer 1. Given that there are four different indicators of IPV, the question is whether greater exposure to drought results in more than one indicator (e.g., control plus physical violence plus emotional violence). This was not addressed by the authors. Second, given the inability to assess duration of exposure to drought with the data available, the authors should note this as a limitation to the study because it weakens the finding of an association between drought and IPV.

*** Reviewer #2: 

The authors have responded to most of the queries comprehensively. However, there is one issue that needs further consideration - the between-country heterogeneity has not been sufficiently addressed in the revision in my view. There is no mention of this major limitation in the discussion. Further, the large heterogeneity would argue against having a pooled estimate in Fig 3 - and ranges can be reported instead. The authors have presented p values for the joint interaction term to estimate the degree of heterogeneity - but it is unusual and I would have expected to see I squared (which are easier to interpret and more widely used). They have argued that they have used a random effects model to account for this - but this is not considered appropriate for observational data if the heterogeneity is substantial.

***

[LINK]

---

## [Editor Report · Decision Letter 2]

21 Feb 2020

Dear Ms. Epstein, 

On behalf of my colleagues and the academic editor, Dr. Lawrence Palinkas, I am delighted to inform you that your manuscript entitled "Drought and intimate partner violence towards women in 19 countries in sub-Saharan Africa during 2011-2018: a population-based study" (PMEDICINE-D-19-03834R2) has been accepted for publication in PLOS Medicine. 

PRODUCTION PROCESS

PRESS

PROFILE INFORMATION

Thank you again for submitting the manuscript to PLOS Medicine. We look forward to publishing it. 

Best wishes, 

Richard Turner, PhD

Senior Editor 

PLOS Medicine

plosmedicine.org